# Proteomics Analysis of Duck Lung Tissues in Response to Highly Pathogenic Avian Influenza Virus

**DOI:** 10.3390/microorganisms12071288

**Published:** 2024-06-25

**Authors:** Periyasamy Vijayakumar, Anamika Mishra, Ram Pratim Deka, Sneha M. Pinto, Yashwanth Subbannayya, Richa Sood, Thottethodi Subrahmanya Keshava Prasad, Ashwin Ashok Raut

**Affiliations:** 1Pathogenomics Laboratory, WOAH Reference Lab for Avian Influenza, ICAR—National Institute of High Security Animal Diseases, Bhopal 462022, Madhya Pradesh, India; nayaganviji@gmail.com (P.V.); reach2anamika@yahoo.com (A.M.); richa.bhatia0609@gmail.com (R.S.); 2Veterinary College and Research Institute, Tamil Nadu Veterinary and Animal Sciences University, Salem 600051, Tamil Nadu, India; 3International Livestock Research Institute, National Agricultural Science Complex, Pusa 110012, New Delhi, India; r.deka@cgiar.org; 4Centre for Systems Biology and Molecular Medicine, Yenepoya (Deemed to be University), Mangalore 575018, Karnataka, India; sneha.mp@gmail.com (S.M.P.); yashwanth.subbannayya@gmail.com (Y.S.); 5School of Biosciences, Faculty of Health and Medical Sciences, University of Surrey, Guildford GU2 7XH, UK

**Keywords:** duck, avian influenza virus, proteomics, phosphoproteome, disease resistance, hub proteins

## Abstract

Domestic ducks (*Anas platyrhynchos domesticus*) are resistant to most of the highly pathogenic avian influenza virus (HPAIV) infections. In this study, we characterized the lung proteome and phosphoproteome of ducks infected with the HPAI H5N1 virus (A/duck/India/02CA10/2011/Agartala) at 12 h, 48 h, and 5 days post-infection. A total of 2082 proteins were differentially expressed and 320 phosphorylation sites mapping to 199 phosphopeptides, corresponding to 129 proteins were identified. The functional annotation of the proteome data analysis revealed the activation of the RIG-I-like receptor and Jak-STAT signaling pathways, which led to the induction of interferon-stimulated gene (ISG) expression. The pathway analysis of the phosphoproteome datasets also confirmed the activation of RIG-I, Jak-STAT signaling, NF-kappa B signaling, and MAPK signaling pathways in the lung tissues. The induction of ISG proteins (STAT1, STAT3, STAT5B, STAT6, IFIT5, and PKR) established a protective anti-viral immune response in duck lung tissue. Further, the protein–protein interaction network analysis identified proteins like AKT1, STAT3, JAK2, RAC1, STAT1, PTPN11, RPS27A, NFKB1, and MAPK1 as the main hub proteins that might play important roles in disease progression in ducks. Together, the functional annotation of the proteome and phosphoproteome datasets revealed the molecular basis of the disease progression and disease resistance mechanism in ducks infected with the HPAI H5N1 virus.

## 1. Introduction

Ducks (*Anas platyrhynchos domesticus*) act as natural reservoir species of avian influenza viruses (AIVs). Most of the highly pathogenic avian influenza virus (HPAIV) infections in domestic ducks are asymptomatic or cause only mild transient clinical signs. This host protection mechanism against influenza infection in ducks may be conferred by increasing innate resistance to infection. The system-level understanding of host innate resistance mechanisms in ducks can help in designing efficient prophylactic and therapeutic strategies against influenza infection [1,2]. High-throughput genomic methods are more suitable tools for the system-level understanding of innate immune responses as they provide global views of the complex innate immune system and reveal the intertwined molecular events that are responsible for disease resistance mechanisms [2]

Transcriptome analysis is presently one of the high-throughput omics technologies that can be easily performed, and these are often used as a proxy for protein abundance. Global host immune gene responses using microarrays/RNA-Seq have been investigated in ducks infected with different AIVs [3,4,5,6,7,8]. These transcriptomics studies revealed that ducks produce an early RIG-I immune response and the expression of cytokines, IFNs, iNOS, STAT3, and IFITM genes [5,6,8,9,10,11,12,13,14,15]. These differential expression patterns of innate immune genes play a crucial role in the disease progression of ducks to AIV infection. Further, the RIG-I immune response is a well-known general concept in ducks for disease resistance; however, an actual explanation of which molecular pathways constitute this innate resistance immune response is lacking.

On the other hand, viral–host interactions are multidimensional in nature, and the transcriptome represents only a single facet of the host response. Other types of high-throughput omic technologies might bring different insights into disease pathogenesis [2,8,16]. The influenza viruses also modulate post-transcriptional regulation and translation, which cannot be addressed by transcriptomics data alone, but requires the integration of proteomic data and other omics profiling data [16,17]. The quantities of mRNAs and the corresponding proteins do not always correlate because proteins are modulated by complex post-transcriptional modification (PTM) processes [18]. Furthermore, PTMs such as phosphorylation, ubiquitination, acetylation, glycosylation, and many others are required for regulating signal transduction and the protein interactions of cellular, biological, and immunological processes [18]. Hence, proteomics analysis, along with phosphoproteomic data, can elucidate the molecular mechanisms of influenza disease progression to a remarkably greater extent. Still, now only a few proteomics studies are available for avian species against avian influenza infection [8,19,20,21]. Further, most current research into innate immune processes against influenza virus infection is biased towards human model systems. Future research work is needed for the wild and domestic aquatic birds because these birds were recognized as the reservoirs of most influenza A viruses [22]. Hence, this study was planned to analyze both proteome and phosphoproteome datasets to further understand the molecular pathways and proteomic determinants responsible for the innate resistant immune mechanisms in ducks infected with the HPAI H5N1 virus.

## 2. Materials and Methods

### 2.1. Experimental Infection of Ducks

Six-week-old healthy domestic ducks, seronegative for AIV, were used for this study. The animal experiments were approved by the Institutional Animal Ethics Committee of ICAR-NIHSAD (approval no. 68/IAEC/HSADL/12 dated 11 May 2012), and all the experiments were conducted in a biosafety level 3 containment facility in the ICAR National Institute of High-Security Animal Diseases, Bhopal, India. The ducks were separated into four groups (n = 5 birds/group). Among the four groups, three groups were intranasally inoculated with 10^6^ EID_50_ of the H5N1 virus (A/duck/India/02CA10/2011/Agartala), and one group (control) was inoculated with PBS. The birds were observed daily for clinical signs. Lung tissues were collected from five birds from each infected group at 12 h, 48 h, and 5 days post-infection. Lung tissues were also collected from the control group at 12 h post-inoculation. The tissues were snap-chilled in liquid nitrogen and stored at −80 °C until protein extraction. The avian influenza virus infection of the lung tissues was confirmed via virus isolation upon inoculation in embryonated chicken eggs (ECEs) and RT-PCR.

### 2.2. Protein Extraction

A total of 150 mg of lung tissue from each sample was washed in 50 mM NH_4_HCO_3_ washing buffer. The lung tissue was cut into small pieces, and 650 μL of SDS protein extraction lysis buffer [0.1% SDS (Invitrogen, Waltham, MA, USA); 50 mM NH_4_HCO_3_ (Sigma, Saint Louis, MO, USA); 1X Complete™ Protease Inhibitor Cocktail (Roche Diagnostics GmbH, Mannheim, Germany)] was added. Tissue samples were homogenized in LZ-Lyser homogenizer at 30 HZ for 2 min. After complete homogenization, the tissue lysate was incubated on ice for 90 min for complete protein lysis. The lysate was centrifuged at 20,000× *g* for 60 min at 4 °C, and the supernatant was collected. The supernatants were immediately snap-heat-treated at 56 °C for 30 min in a dry bath for the inactivation of HPAIV H5N1 in the protein extracts. All the heat-treated samples were stored at −80 °C for mass spectrometry analysis.

### 2.3. Sample Preparation for LC-MS Analysis

The protein amount in each sample was estimated using the Bicinchoninic Acid (BCA) assay. The quality of the lung proteins was checked using SDS-PAGE. A pool for each time point was prepared by pooling 150 μg of the protein lysate each from the 3 best samples at that time point. Protein samples were reduced for 20 min at 60 °C in 10 mM dithiothreitol solution, followed by alkylation for 10 min in 20 mM iodoacetamide in the dark at room temperature. To remove the SDS from the samples, the samples were subjected to acetone precipitation (6X). The protein pellet obtained from each condition was then centrifuged at 13,000× *g* for 10 min at 4 °C to remove the acetone and impurities. The pellets were then resuspended in 100 mM TEABC buffer and subjected to trypsin digestion (enzyme: substrate 1:20) at 37 °C overnight. After the trypsin digestion, the samples were subjected to Tandem Mass Tag™ labeling (TMT) labeling (Thermo Scientific™, Waltham, MA, USA). The samples were labeled with the following TMT channels: control: 127C; D12H–128N; D48H–129C; D5D–130N, as per the manufacturer’s protocol. After the TMT labeling, a 5 µL labeled sample was taken from each condition and pooled, and the pooled sample was subjected to high pH reversed-phase liquid chromatography separation. A total of 96 fractions were collected which were subsequently pooled to 12 fractions. Then, 1/10th of the volume from these was transferred to Eppendorf tubes and used for LC-MS/MS analysis in an Orbitrap Fusion Tribrid mass spectrometer (Thermo Scientific™) to determine the expression of the total quantitative proteome. The remaining samples were pooled to obtain 6 fractions, which were subjected to phosphoproteomics analysis. Each fraction was subjected to phosphopeptide enrichment using TiO_2_ beads. Briefly, the peptide samples were incubated with TiO_2_ beads in a ratio of 1:10 for 15 min, followed by washing the bound beads and eluting the bound phosphopeptides. LC-MS/MS analysis was carried out using the Orbitrap Fusion Tribrid mass spectrometer to determine the extent of phosphorylation across various time points in the enriched fractions. The peptides fragmented via HCD fragmentation with a collision energy of 34. The data were acquired in the data-dependent mode (DDA) with the top 10 MS2.

### 2.4. Bioinformatics Analysis

The data obtained from both the total proteome and phosphoproteome datasets were analyzed in the Proteome Discoverer 2.1 software suite (Thermo Scientific™). The data were searched using the SequestHT algorithm. The reference duck proteome dataset was downloaded from the NCBI database. The important parameters used for the database search were as follows: peptide mass error tolerance level: 10 ppm, fragment mass error tolerance level: 0.05 Da, and the number of missed cleavages: 2, and carbamidomethylation of cysteine, TMT label at N-term of peptides, and lysine residues were set at fixed modification, and the oxidation of methionine was set at variable modification. In addition to the above parameters, for the identification of phosphoproteins, phosphorylation at serine (S), threonine (T), and tyrosine (Y) was set at variable modification. The relative ratio for each phosphopeptide was calculated by dividing the intensity of each phosphorylated peptide by the intensity of the corresponding peptide. The highly confident PTM sites were filtered using a PhosphoRS score above 75%. The data were searched against the target decoy database and the false discovery rate was set to 1% at the protein and peptide level. The functional classification of the proteins and phosphoproteins was performed for gene ontology (GO) in the database for annotation, visualization, and integrated discovery (DAVID) [23] and pathway analysis in the NetworkAnalyst web server using the Kyoto encyclopedia of genes and genomes (KEGGs) database [24,25]. Featuring a web server and web service for the functional annotation and enrichment analyses of gene lists, DAVID is a well-known bioinformatics resource system. We used the online web server NetworkAnalyst for the construction of the protein–protein interaction (PPI) networks [24]. The main driving or hub proteins were identified based on two topological measures, degree centrality and betweenness centrality. The heatmap was generated using the “Clustvis” web tool [26].

## 3. Results and Discussion

### 3.1. Differential Protein Expression Analysis

The duck lung raw proteome dataset contained a total of 120,000 MS spectra and 30,000 MS/MS spectra. The 30,000 MS/MS spectra were searched against the NCBI duck proteome database. A total of 2082 proteins were differentially expressed in duck lung tissues infected with the HPAI H5N1 virus at the 1% FDR level. The analysis of the differential expression of proteins at 12 h intervals showed that 876 proteins were upregulated, and 1179 proteins were downregulated. A total of 898 proteins were upregulated, and 1159 proteins were downregulated at 48 h post-infection. At 5 days post-infection, 1095 proteins were upregulated, and 962 proteins were downregulated in duck lung tissues (Table 1). The protein profile showed that 113, 72, and 205 proteins were exclusively upregulated in H5N1-infected duck lung tissue at 12 h, 48 h, and 5 days post-infection, respectively (Figure 1). The downregulation of 154, 112, and 49 proteins was observed exclusively at 12 h, 48 h, and 5 days post-infection, respectively (Figure 1). The fold change value of the upregulated proteins ranged from 11.8 to 1. Interestingly, at the 12 and 48 h time points, a higher number of proteins were downregulated as compared to the 5 day time point post-infection condition (Table 1). The heatmap of the differentially expressed proteins of HPAIV-infected duck lung tissues is represented in Figure 2. This result indicates that many of the host proteins were downregulated at the initial stage of infection, and then later, the host adjusted the cellular homeostasis to recover from the virus-induced differential expression of proteins.

The gene ontology (GO) analysis of the exclusively upregulated and downregulated lung tissue proteins of H5N1-infected ducks was achieved using DAVID 2021 tools. This analysis result provides information on how the disease progresses from the commencement of the disease to the end of disease progression in ducks infected with the HPAI H5N1 virus. At 12 h intervals, the upregulated proteins enriched the Fc-epsilon receptor signaling pathway, MAPK cascade, Wnt signaling pathway, antigen processing and presentation of exogenous peptide antigen via MHC class I, NIK/NF-kappaB signaling, tumor necrosis factor-mediated signaling pathway, endocytosis, and small GTPase, which mediated the signal transduction GO terms. The downregulated proteins enriched the ATP binding, the glycolytic process, protein refolding, the extracellular matrix, the intracellular ribonucleoprotein complex, gluconeogenesis, NAD binding, translational initiation, mRNA transport, and protein transport GO terms related to cellular homeostasis (Figure 3). These patterns indicate that, at 12 h time intervals, virus–host interactions resulted in the activation of host signaling pathways and inhibited the cellular homeostasis process. At the 48 h post-infection stage, the upregulated proteins enriched the movement of cell components, mRNA splicing, focal adhesion, the actin cytoskeleton, and the spliceosomal complex, and the downregulated proteins enriched the Golgi membrane, endoplasmic reticulum lumen, lysosome, and calcium ion transport GO terms (Figure 4). At the 5-day post-infection stage, most of the cellular homeostasis processes like the regulation of mRNA stability, protein polyubiquitination, cytoskeleton organization, translational initiation, the tricarboxylic acid cycle, and organelle assembly, and the positive regulation of the cellular protein catabolic process, transport vesicle, GTPase activity, and the kinase activity GO terms were enriched (Figure 5). In summary, these results indicate that the inhibition of cellular homeostasis and activation of signaling pathways occurs at the initial stage of infection, and the host resumes normal cellular homeostasis at the later stage of infection.

### 3.2. Differential Phosphoproteomic Expression Analysis

In this study, we investigated the phosphorylation-regulated host proteins and signaling pathways of HPAI H5N1-infected duck lung tissues at different time points post-infection. We identified 320 phosphorylation sites mapping to 199 phosphopeptides, corresponding to 129 proteins. In all, 93, 93, and 104 phosphosites were differentially phosphorylated (1.5-fold up and down) at 12 h, 48 h, and 5 days post-infection, respectively, in duck lung tissues. The phosphoproteome profile showed that 36, 27, and 40 sites were hyperphosphorylated in duck lung tissue proteins at 12 h, 48 h, and 5 days post-infection, respectively. Further, 57, 66, and 64 sites were observed as hypophosphorylated in the lung tissue proteins at 12 h, 48 h, and 5 days post-infection, respectively. Post-translational modifications (PTMs) are crucial regulatory cellular processes in which proteins are enzymatically modified to reversibly modulate the activity, subcellular localization, conformation, and/or protein–protein interactions of the target protein. Thus, PTMs are essential for the dynamic regulation of intracellular pathogen-sensing signal transduction pathways [27]. In RIG-I and interferon-mediated signaling pathways, ubiquitination and phosphorylation events are important for the induction of anti-viral innate immune responses. Phosphorylation is the most important PTM, which involves the transfer of a phosphate group by a protein kinase to a serine (S), threonine (T), and tyrosine (Y) residue of a target protein substrate.

Our phosphoproteome profile indicates that the HPAI H5N1 virus has a substantial impact on host protein phosphorylation. Most proteins were phosphorylated at the Ser residues in comparison with the Thr and Tyr residues, which agrees with a previous study [28,29]. To elucidate the kinase family involved in influenza infection conditions, we used a group-based prediction system (GPS) [30]. The GPS ranks the likelihood that a particular kinase or kinase family phosphorylates a given phosphorylation site by considering the amino acids surrounding the phosphorylation site. The AGC, CK1, CMGC, and CAMK protein kinase families were predicted to be activated during HPAI H5N1 infection conditions in ducks. The activation of these protein kinase family members in influenza infection was reported in a previous study [28,29,31].

### 3.3. Pathway Analysis of Proteomics Datasets

In domestic ducks, most HPAI virus infections cause no or mild clinical signs and lesions [32,33]. Understanding the disease-resistance immune responses in ducks in HPAIV infection may provide key insights into the immune pathways required for prophylactic or therapeutic protection. Previous literature specifies that HPAI virus-infected ducks produce an early protective type I interferon response and thereby recover from influenza-induced inflammatory pathology [5,6,14,34,35]. However, the relationship between type I interferon and inflammation is a complex and multifactorial cellular process [36]. Further, the several signaling pathways activated during the type I interferon response are yet to be confirmed at the protein and phosphoproteome levels in ducks. Hence, the proteome and phosphoproteome datasets were used for pathway analysis to identify the signaling pathways that are activated in ducks during HPAIV infection. The molecular pathway analysis of differentially expressed proteins showed the activation of the RIG-I-like receptor signaling pathway, Jak-STAT signaling pathway, PI3K-Akt signaling pathway, MAPK signaling pathway, NOD-like receptor signaling pathway, and Toll-like receptor signaling pathways in the lung tissue of ducks (Table 2).

### 3.4. RIG-I-Like Receptor Signaling Pathway

Retinoic acid-inducible gene I is thought to be an important pattern-recognizing receptor (PRR) during influenza infection [37]. The RLR family includes three members: RIG-I, melanoma differentiation-associated gene 5 (MDA5), and laboratory of genetics and physiology 2 (LGP2) [38]. We observed the upregulation of the expression of the MDA5 and LGP2 proteins in the duck H5N1-infected lung tissues. The significant upregulation of MDA5 and LGP2 has been observed in the lung tissues of Muscovy ducks infected with HPAI H5N1 [39,40]. In ducks, both MDA5 and LGP2 activate IRF-7-dependent signaling pathways and induce interferons (IFNs) as well as interferon-stimulated gene (ISG) production, both of which mediate the anti-viral and pro-inflammatory responses during HPAI H5N1 virus infection [8,39,40,41,42,43,44]. The tripartite motif (TRIM) family of proteins, tripartite motif 25 (TRIM25), and ubiquitin carboxyl-terminal hydrolase 15 (USP15) were expressed in the duck lung tissues (Figure 6). The IFN-inducible E3 ubiquitin ligase TRIM25 protein is a key regulator of the RIG-I-mediated IFN response and modifies RIG-I with K63-linked polyubiquitination [43,45]. The CARD domains of RIG-I and MDA5 were ubiquitinated by TRIM25 [46,47,48]. TRIM25 stabilizes the RIG-I–2CARD: MAVS–CARD helical structure by furnishing the short chains of K63-linked ubiquitin molecules and amplifying MAVS signaling [45]. Further, duck TRIM25 provides both anchored and unanchored K63-linked polyubiquitin chains to the CARD domains of RIG-I [48]. USP15 is a deubiquitinase, which removes K48-linked ubiquitination from the TRIM25 SPRY domain, thus stabilizing TRIM25 and inducing a sustained cytokine response. Taken together, RIG-I-like receptor ubiquitination by the TRIM25 protein induces RIG-I oligomerization, and its interaction with MAVS activates the RIG-I signaling pathway and induces anti-viral gene expression in the lung tissues of HPAI H5N1 virus-infected ducks.

### 3.5. Interferon Signaling: The Jak-Stat Pathway

Janus kinase 2 (JAK2) is ubiquitously expressed and binds to the heterodimers of the IFN-γ receptor 1 (IFNGR1) and 2 (IFNGR2) chains on the inner side of the membrane. JAKs provide receptors with stability, facilitate their cell-surface localization, and serve as key components of IFN signaling complexes [49,50,51]. The upregulation of the JAK2 protein in all three post-infection conditions was observed in the HPAIV-infected duck lung tissues (Figure 6). Upon binding type II IFN to IFNGR1/2, the receptor complex leads to the phosphorylation of preassociated JAK1 and JAK2 tyrosine kinases, and the transphosphorylation of the receptor chains leads to the recruitment and phosphorylation of the signal transducers and activators of transcription 1 (STAT1) [52]. The homodimers of phosphorylated STAT1 proteins form the IFN-γ activation factor (GAF). The GAF translocates into the nucleus and binds to the gamma-activated sequence promoter elements, resulting in the expression of ISGs [53].

STAT proteins have the dual function of signal transduction and the activation of transcription [54]. In the duck proteome dataset, the expression of STAT proteins like STAT1, STAT3, STAT5B, and STAT6 were observed (Figure 6). It has also previously been reported that the influenza A virus induces the expression of the STAT protein in the lung tissues of ducks [6,13,55]. STAT1 is critical in signal transduction from the type I IFNs and the type II IFNs [52,56,57]. Previous studies showed that SOCS-3 expression negatively affects STAT phosphorylation [58,59]. The STAT3 protein acts as a transcription factor, plays a critical role in the IFN signaling pathways, and is required for a robust IFN-induced anti-viral response [60]. The STAT3 protein has an antagonistic effect on the inflammatory cytokine response and promotes a strong anti-inflammatory response [6,13,55]. STAT5B is activated in response to a variety of cytokines, binds to the gamma interferon activation site element, and activates prolactin-induced transcription [61,62].

The interferon-induced protein with tetratricopeptide 5 (IFIT5) exhibited anti-viral activity against the influenza virus by sequestering single-strand viral RNA with 5’triphosphate [63]. The IFIT5 protein, at 12 h intervals, was downregulated, and in later post-infection conditions, IFIT5 showed upregulation to counter viral infection. Similar kinds of upregulation were observed in duck lungs and spleens in response to HPAI H5N1 viruses [6,35,64,65]. PKR is an interferon-induced protein with anti-viral, anti-proliferative, and pro-apoptotic functions. PKR is upregulated by the type I and type III IFN signaling pathways [66]. PKR has a known anti-influenza A virus effect by phosphorylating the alpha subunit of eukaryotic initiation factor 2 (EIF2α), which shuts down cellular and influenza A viral protein synthesis, thereby effectively reducing viral replication [67,68]. PKR is highly upregulated in the lungs of HPAI-infected ducks, and the same pattern has been reported (Figure 6) [6,65]. In summary, our datasets showed evidence that protective interferon responses were activated through the stimulation of RIG-I-like receptor signaling and the Jak-STAT signaling pathways, which resulted in the induction of the expression of ISGs (STAT1, STAT3, STAT5B, STAT6, IFIT5, and PKR).

### 3.6. Pathway Analysis of Phosphoproteomics Datasets

To further gain insight into the activation of specific signal transduction pathways by HPAI H5N1 infection conditions in ducks, we analyzed the phosphoproteome data using the KEGG pathway database (Table 3). The most notable cellular pathways activated for differentially regulated phosphoproteins in duck lung tissues included the Jak-STAT signaling pathway, NF-kappa B signaling pathway, MAPK signaling pathway, ErbB signaling pathway, PI3K-Akt signaling pathway, Rap1 signaling pathway, regulation of the actin cytoskeleton, prolactin signaling pathway, chemokine signaling pathway, T cell receptor signaling pathway, endocytosis, tight junctions, etc. Among these pathways, the Jak-STAT signaling pathway, NF-kappa B signaling pathway, and MAPK signaling pathway were interrelated with RIG-I and the interferon-mediated signaling pathways. Hence, our phosphoproteome dataset pathway analysis also confirmed the activation of RIG-I and interferon-mediated signaling pathways in HPAIV infection conditions in ducks. The activation of the MAPK signaling pathway, endocytosis, and tight junctions in influenza virus infection conditions was reported [29]. Furthermore, the activation of focal adhesions and the actin cytoskeleton pathway involved in the early stages of IAV infection was also reported [29,31]. Lakadamyali et al., 2003 suggested the endocytic pathway toward late endosomes: endosome maturation and initial acidification occur in the perinuclear region [69]. Sun and Whittaker (2007) reported the indispensable roles of a dynamic actin cytoskeleton for influenza virus entry into the epithelial cells [70]. Thus, the activation of these cellular functions is essential for influenza virus entry into host cells.

Influenza viruses can hijack the various intracellular signaling pathways to support their efficient replication. Among the signaling pathways, the activation of PI3K/Akt signaling, the MAPK (Raf/MEK/ERK) signaling pathway, and the NF-κB signaling pathways were reported as essential pathways responsible for virus-supportive replication. The influenza NS1 protein binds to the SH2 domain of the p85 sub8unit and activates the PI3K/Akt signaling pathways [71,72]. The activation of this pathway prevents premature apoptosis, thereby promoting efficient virus replication at a late step of the infection [22,73]. MAPK (Raf/MEK/ERK) signaling pathways encompass cascades of kinases that convert extracellular signals into cellular responses [74]. The activation of the Raf/MEK/ERK signaling pathway facilitates influenza-supportive virus replication roles by inducing the nuclear export of viral ribonucleoprotein complexes (RNPs) at the late stages of the viral life cycle [75]. Further, influenza virus infection impairs the Ras-Raf-MEK-ERK pathway by downregulating MEK1 SUMOylation to facilitate viral RNP export and virus propagation [76]. The activation of the NF κB signaling pathway is one of the essential cellular responses regulating anti-viral cytokine and interferon-β expression [77]. Some previous studies have suggested that influenza viruses-elicited NF-κB activity, which helps in virus replication and spreading and NF-κB inhibition, also impairs IAV propagation [78,79,80]. In a nutshell, phosphoproteome dataset analysis provides insights into multiple kinase-mediated signaling pathways that were activated in HPAI H5N1 virus-infected duck lung tissues.

### 3.7. Identification of Proteomic Determinants

Biological network analysis is a powerful approach to gain a systems-level understanding of disease pathogenesis. To know the main driving or hub proteins responsible for disease resistance to HPAIV infection conditions in ducks, we constructed protein–protein interaction (PPI) networks for the differentially expressed lung proteins of ducks. Among different molecular networks, PPI networks have emerged as an important resource because protein interactions play fundamental roles in structuring and mediating essentially all biological processes. PPI networks are often presented as undirected graphs with nodes as proteins and edges indicating interactions between two connecting proteins.

Proteins such as AKT1, STAT3, JAK2, RAC1, STAT1, PTPN11, RPS27A, NFKB1, MAPK1, etc., were identified with high degree centrality and betweenness centrality measures as the main hub proteins responsible for disease pathogenesis in ducks (Table 4) (Figure 7). The functional proteomics and phosphoproteomics analysis of our datasets showed that the activation of RIG-I signaling and Jak-STAT signaling pathways was responsible for disease resistance against HPAI H5N1 virus infection in ducks. The JAK2, STAT1, STAT3, and NFKB1 proteins were involved in these pathways and were identified as the main hub proteins for disease pathogenesis in ducks. These proteins act as the main signaling molecules for the induction of the expression of ISGs and the establishment of anti-viral states in the host cells.

Akt (protein kinase B, PKB) is a serine/threonine kinase, and Akt is activated by the lipid products of phosphoinositide 3′-kinase (PI3K). Upon activation, Akt regulates the activities related to proliferation, the cell cycle, glycogen synthesis, angiogenesis, and telomerase [81]. In particular, the PI3K–Akt pathway plays a critical role in the uptake of a virus during viral entry, the prevention of premature apoptosis, and viral RNA expression and RNP localization [72,73,82]. RAC1 belongs to the family of Rho GTPases that regulate a wide variety of cellular processes. RAC1 is reported to have virus-supportive as well as virus-suppressive functions for different viruses [83,84,85,86]. In the case of the influenza virus infections condition, RAC1 has an anti-viral role by inducing interferon-β production and a crucial virus-supportive role in the activity of the viral polymerase complex [87,88]. The influenza A virus (IAV) blocks RAC1-mediated host cell signal transduction through the NS1 protein to facilitate its replication [89]. Further, RAC1 is proposed as a cellular target for the therapeutic treatment of influenza virus infections, as the inhibition of RAC1 by the small chemical compound NSC23766 resulted in the impaired replication of a wide variety of influenza viruses [88].

Protein tyrosine phosphatase non-receptor type 11 (PTPN11) is a member of the protein tyrosine phosphatase family. The expression of PTPN11 genes has been reported in influenza infection conditions [90,91]. SHP-2, the protein encoded by PTPN11, interacts with the transcriptional activator STAT3 [92]. PTPN11 interacts with long intergenic non-coding RNA LINC00673, which promotes cell growth and proliferation by activating SRC-ERK signaling and inhibiting STAT1 signaling [93]. PTPN11 positively regulates the MAPK signal transduction pathway [94]. NFKB1 is a subunit of the NF-kappaB (NF-κB) protein complex. The activation of NF-κB plays an important role in the inhibition of virus replication by stimulating the synthesis of IFN-α/β, which leads to the expression of anti-viral genes [95,96]. However, the NS1 protein of the influenza A virus can prevent NF-κB activation, thereby subverting the IFN innate immune system [97].

MAPK1 is a member of the MAP kinase family. MAPKs play critical roles in regulating the expression of various pro-inflammatory cytokines, including IL-6 and TNF-alpha, in response to microbial infection [98]. The activation of p38 mitogen-activated protein kinase (p38 MAPK) induces the dysregulation of cytokine expression in HPAIV-infected primary human monocyte-derived macrophages and bronchial epithelial cells [99,100,101]. Further, the inhibition of p38 MAPK significantly reduces the hyperinduction of cytokines and prevents cytokine-induced pathogenicity in HPAIV-infected mice and human macrophages [99,102]. Therefore, interference with the p38 MAPK pathway is also proposed for therapeutic treatments for HPAIV infection [102]. The RPS27A and UBA52 genes code for ubiquitin, and this ubiquitin is a highly conserved protein that has a major role in targeting cellular proteins for degradation in the 26S proteasome [103,104]. Further, the UBA52 host protein interacts with the viral RNA polymerase acidic protein (PA), PA-N155, and PA-N182 and promotes the replication of the highly pathogenic H5N1 avian influenza virus [105].

The Ras homolog gene family member A (RhoA) is a small GTPase that controls gene transcription, actin polymerization, cell cycle progression, and cell transformation. The influenza virus NS1 protein induces G_0_/G_1_ cell cycle arrest mainly by interfering with the RhoA/pRb signaling cascade [106]. In summary, we identified many proteomic determinants involved in the RIG-I, Jak-STAT, and other signaling pathways responsible for disease progression and disease resistance in ducks infected with HPAIV using PPI network analysis. Most of the proteomic determinants identified in this study were previously reported in human model systems and showed evidence of playing a very essential role in disease pathogenesis in influenza infection conditions. Here, we identified the expression of these proteins in duck lung tissues and functionally annotated their role in disease progression. Further, some of these proteomic determinants were already tested as novel cellular targets for the therapeutic treatment of influenza virus infections. However, further in vivo studies are required to validate the role of proteomic determinants in disease progression and find novel host proteins as a therapeutic or prophylactic treatment in HPAIV infection in ducks.

## 4. Conclusions

In conclusion, the bioinformatics analysis of the proteome and phosphoproteome datasets revealed the activation of RIG-I-like receptor signaling and the Jak-STAT signaling pathways, which resulted in the expression of ISG proteins (STAT1, STAT3, STAT5B, STAT6, IFIT5, and PKR), thus establishing a protective anti-viral immune response in HPAI H5N1 virus-infected lung tissues. The activation of these pathways and expression of interferon-stimulated proteins may restrict viral replication and recovery from influenza virus-induced inflammatory changes. Therefore, ducks may develop disease resistance against the HPAI H5N1 virus infection condition. Through PPI network analysis, we identified many proteomic determinants (AKT1, STAT3, JAK2, RAC1, STAT1, PTPN11, RPS27A, NFKB1, and MAPK1) that might play important roles in disease progression and disease resistance in ducks.

## Figures and Tables

**Figure 1 microorganisms-12-01288-f001:**
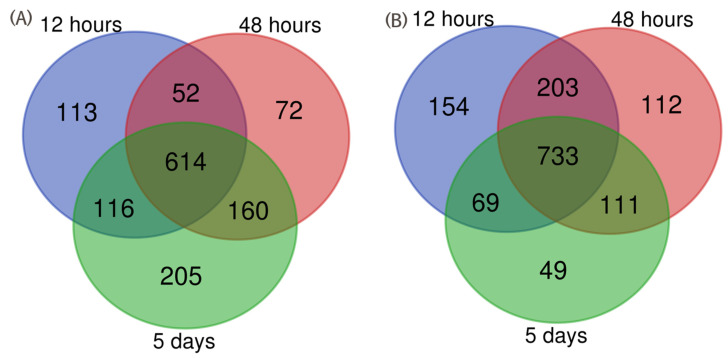
Comparative analysis of upregulated (**A**) and downregulated (**B**) protein expression changes at different time intervals in duck lung tissues.

**Figure 2 microorganisms-12-01288-f002:**
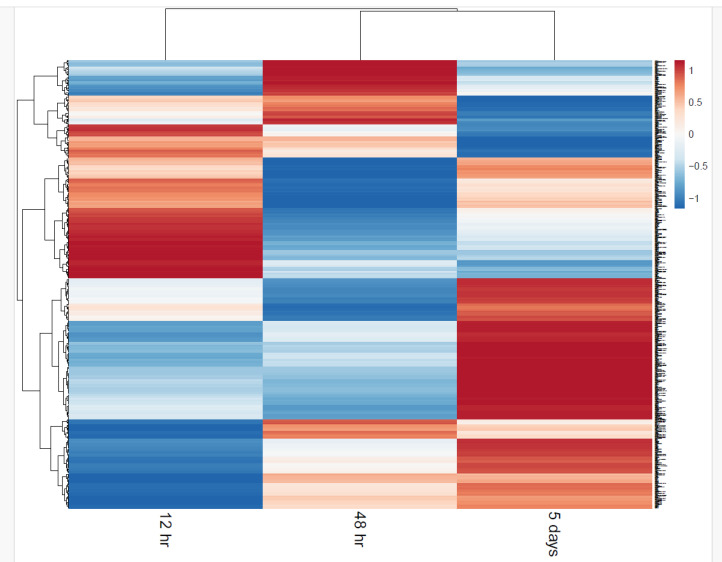
Heatmap of differentially expressed proteins of HPAIV-infected lung tissues. The expression levels are visualized using a gradient color scheme.

**Figure 3 microorganisms-12-01288-f003:**
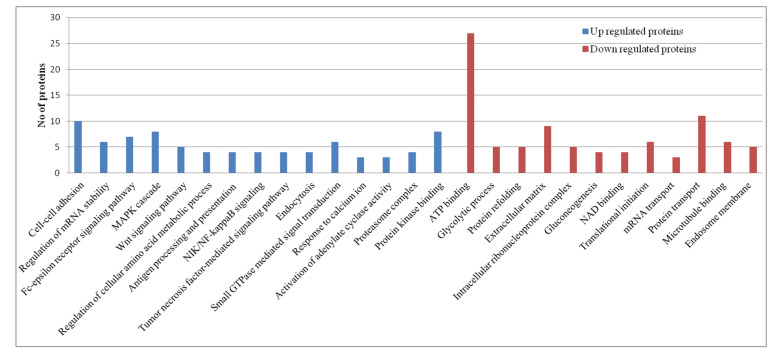
Gene ontology term analysis of the exclusively upregulated and downregulated proteins in duck lung tissues infected with the HPAI H5N1 virus at 12 h post-infection. The GO terms were selected based on a cut-off *p* value > 0.05.

**Figure 4 microorganisms-12-01288-f004:**
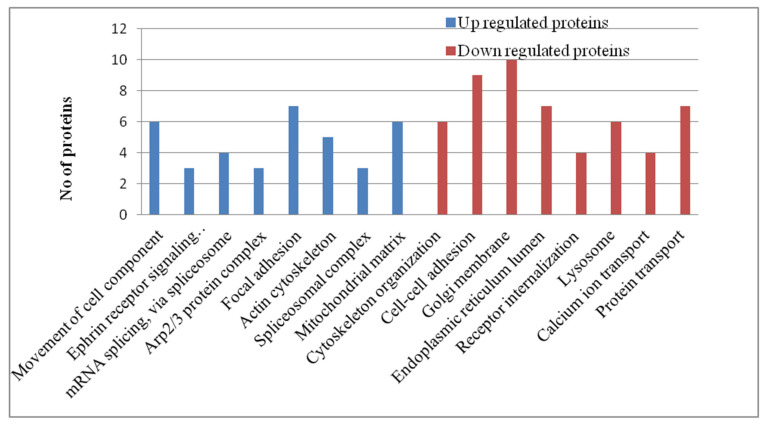
Gene ontology term analysis of the exclusively upregulated and downregulated proteins in duck lung tissues infected with the HPAI H5N1 virus at 48 h post-infection. The GO terms were selected based on a cut-off *p* value > 0.05.

**Figure 5 microorganisms-12-01288-f005:**
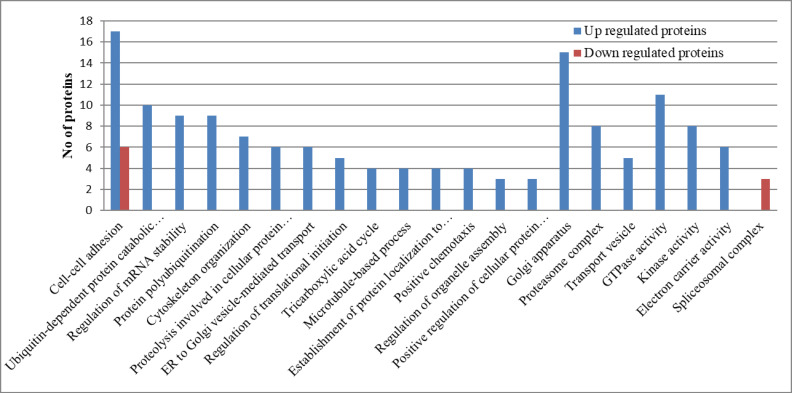
Gene ontology term analysis of the exclusively upregulated and downregulated proteins in duck lung tissues infected with the HPAI H5N1 virus at 5 days post-infection. The GO terms were selected based on a cut-off *p* value > 0.05.

**Figure 6 microorganisms-12-01288-f006:**
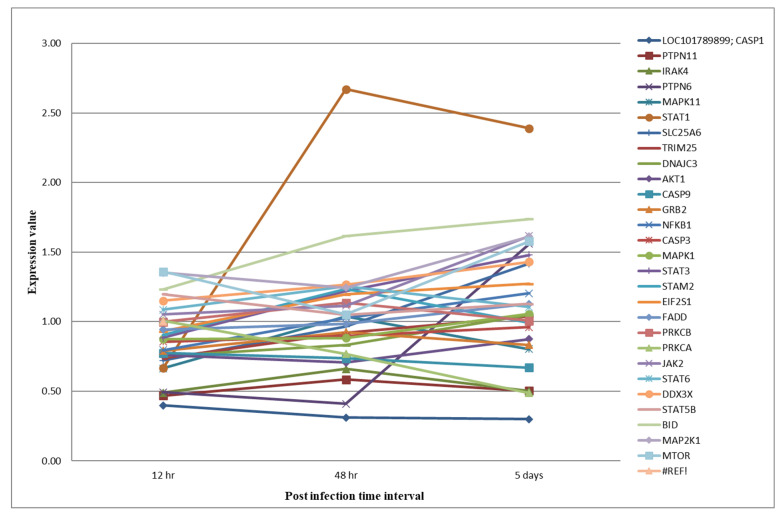
Expression levels of proteins involved in the RIG-I-like receptor and Jak-STAT signaling pathways in duck lung tissues infected with HPAIVs.

**Figure 7 microorganisms-12-01288-f007:**
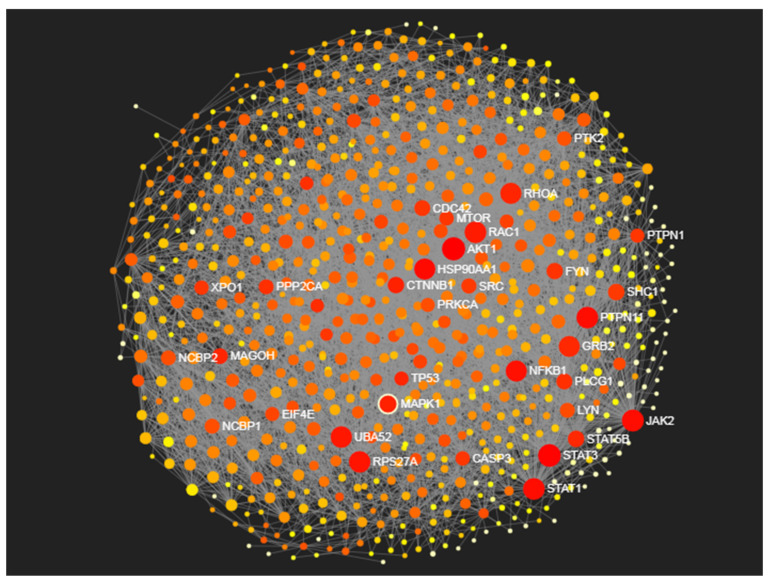
The protein–protein interaction network of duck lung tissues infected with the HPAI H5N1 virus. The important hub proteins involved in influenza pathogenesis are highlighted in red color in the PPI network.

**Table 1 microorganisms-12-01288-t001:** Differential protein expression analysis in ducks infected with the HPAI H5N1 virus.

Condition	Upregulated Proteins	Downregulated Proteins	Upregulated Proteins (>1.5 Fold)	Downregulated Proteins (<1.5 Fold)
12 h interval	895	1159	128	225
48 h interval	898	1159	199	277
5 day interval	1095	962	296	264

**Table 2 microorganisms-12-01288-t002:** KEGG pathway analysis of differentially expressed proteins in duck lung tissues infected with the HPAI H5N1 virus.

Pathway	Hits	FDR
Focal adhesion	179	3.20 × 10^−21^
PI3K-Akt signaling pathway	291	1.10 × 10^−20^
MAPK signaling pathway	243	1.12 × 10^−17^
Cell cycle	116	1.97 × 10^−17^
Endocytosis	204	2.00 × 10^−16^
AMPK signaling pathway	110	5.46 × 10^−15^
Regulation of the actin cytoskeleton	179	1.36 × 10^−14^
Phospholipase D signaling pathway	130	3.91 × 10^−14^
T cell receptor signaling pathway	94	4.54 × 10^−14^
Chemokine signaling pathway	159	3.77 × 10^−13^
Apoptosis	119	8.99 × 10^−13^
Spliceosome	117	1.91 × 10^−12^
Platelet activation	109	4.49 × 10^−12^
Ubiquitin-mediated proteolysis	117	4.00 × 10^−11^
mTOR signaling pathway	128	8.53 × 10^−11^
Fc epsilon RI signaling pathway	64	1.68 × 10^−10^
Influenza A	137	2.43 × 10^−10^
Jak-STAT signaling pathway	132	1.20 × 10^−9^
Protein processing in the endoplasmic reticulum	134	1.32 × 10^−9^
Proteasome	44	4.35 × 10^−9^
Toll-like receptor signaling pathway	88	2.57 × 10^−8^
RIG-I-like receptor signaling pathway	62	1.09 × 10^−7^
Metabolic pathways	926	1.26 × 10^−7^
TNF signaling pathway	91	1.26 × 10^−7^
Th1 and Th2 cell differentiation	77	4.78 × 10^−7^
Leukocyte transendothelial migration	91	5.95 × 10^−7^

**Table 3 microorganisms-12-01288-t003:** KEGG pathway analysis of differentially expressed phosphoproteins in duck lung tissues infected with the HPAI H5N1 virus.

Pathway	Hits	*p* Value
Jak-STAT signaling pathway	5	0.0191
NF-kappa B signaling pathway	5	0.0026
MAPK signaling pathway	14	5.84 × 10^−7^
PI3K-Akt signaling pathway	22	7.33 × 10^−13^
Focal adhesion	34	2.84 × 10^−35^
Regulation of the actin cytoskeleton	24	5.77 × 10^−20^
ErbB signaling pathway	14	3.30 × 10^−14^
Leukocyte transendothelial migration	14	1.71 × 10^−12^
Ras signaling pathway	16	3.65 × 10^−10^
Rap1 signaling pathway	15	6.58 × 10^−10^
T cell receptor signaling pathway	10	3.29 × 10^−8^
Chemokine signaling pathway	12	1.97 × 10^−7^
Phospholipase D signaling pathway	10	1.22 × 10^−6^
Tight junction	9	3.13 × 10^−5^
Endocytosis	8	0.00218
Prolactin signaling pathway	4	0.00439

**Table 4 microorganisms-12-01288-t004:** Hub proteins identified in duck PPI networks based on degree centrality and betweenness centrality measures.

Proteins	Degree Centrality	Betweenness Centrality
AKT1	151	23,549.86
STAT3	138	20,538.68
JAK2	122	14,684.79
RAC1	117	9430.98
STAT1	116	14,916.66
PTPN11	116	14,221.68
RPS27A	113	11,935.64
UBA52	110	11,175.27
HSP90AA1	107	17,096.24
RHOA	107	7556.94
NFKB1	106	13,320.84
GRB2	103	6744.12
MAPK1	100	8118.52

## Data Availability

The raw data supporting the conclusions of this article will be made available by the authors on request.

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
