# Peer review of "Proteomics Analysis of Duck Lung Tissues in Response to Highly Pathogenic Avian Influenza Virus"

_microorganisms, 2024, doi:10.3390/microorganisms12071288_

Round 1

Reviewer 1 Report

Comments and Suggestions for Authors

In their article ” Proteomics analysis of duck lung tissues in response to highly pathogenic avian influenza virus “ authors present a study, where they characterized the lung proteome and phosphoproteome of ducks infected with HPAI H5N1 virus (A/duck/In dia/02CA10/2011/Agartala) at 12 hr, 48 hr and 5 days post infection condition.

The aim of the study is to present the role of proteome and phosphoproteome datasets reveals the molecular basis of the disease progression and disease resistance mechanism in ducks infected with HPAI H5N1 virus. Every new knowledge concerning HPAI is very important, because Influenza A viruses is still a potential danger for human and animal health.

I think the manuscript is interesting, but the font, figure size and sub-headings need to be fixed first.

Here are some additional notes:

Materials and Methods.

Line 86 - “Lung tissues were also collected from the control group at 12hr post inoculation” My question is: Are all the animals from control group were tested only at 12hr post inoculation?

Line 111 - Tandem Mass Tag™ - write the name of the manufacturer

Line 123 - Orbitrap Fusion Tribrid mass spectrometer - write the name of the manufacturer

Line 130 - Proteome Discoverer 2.1 software- write the name of the manufacturer

Results.

Line 179 – “DAVID tools” you need to explain this tool in section Materials and Methods.

Lines 215-222; 241-250 The paragraphs are not proper for this part of the manuscript. First you have to explain your results and then you can compare them with other studies.

Reviewer 2 Report

Comments and Suggestions for Authors

First, this paper does not provide proper reference citations. Therefore, there is no accurate comparative verification with similar previous experimental results. In particular, Ye at al. reported the differential proteome response to HPAI H5N1 viruses infection in Muscovy duck in 2022 (Front. Immunol, Volume 13). As both the domestic duck and the Muscovy duck belong to the Anatidae family, the differences and similarities between the results in this paper and Ye's results must be discussed by the authors. Related to this mention, the Results and Discussion section simply lists many facts from previous studies. It should clearly state what new knowledge has been gained from the proteome and phosphoproteome analysis of the current study. Second, in the Introduction, the authors describe that they conducted this study to understand the molecular pathways and proteomic determinants responsible for the "innate resistant immune mechanisms" in ducks infected with HPAI H5N1 virus. However, they do not discuss the possibility of which molecular pathways or proteomic determinants may play a critical role in the asymptomatic or mildly ill ducks. To achieve their purpose, I recommend that they consider their findings more deeply. Third, in the Materials & Methods, the authors described that avian influenza virus infection of lung tissues was confirmed by virus isolation upon inoculation in embryonated chicken eggs and RT-PCR (Line 88-89). However, there is no result in the paper. Changes in the viral load over the course of the observation period (12 hpi, 48 hpi and 5 dpi) should be noted in the results, as they may be closely related to the response of the host. It would be better to summarize the results of virus isolation and PT-PCR in a table. Fourth, regarding the Interferon signaling section (line 286-325) in the Results & Discussion, I often could not find the results of the expression level of the proteins in Figure 6. Which line in Figure 6 shows the expression change of JAK2? I have the same question for STAT5B and STAT6 (line 301), IFIT5 (line 313) and PKR (line 320)

Comments on the Quality of English Language

I found their work of interest. However, there are some critical issues that should be addressed before publication. 

Reviewer 3 Report

Comments and Suggestions for Authors

The article addresses a topic of interest. However, there are some aspects to refine, before considering the manuscript for publication:

1) presence of some acronyms not explicitly stated (e.g. BCA, line 102);

2) I would suggest combining the data reported in Figures 3-5 into a single Figure to facilitate the analysis of the dynamics occurring at different infection times. Since the results are reported at different scales in the three above-mentioned figures, their analysis is currently difficult;

3) the text of the paragraph “Differential phosphoproteomic …” is in italics and the data described are apparently not illustrated in any Figure/Table. I suggest reporting in the paragraph that “data are not shown” and justifying this choice;

4) apparently in Figure 6 the following result is not reported: “Up-regulation of JAK2 protein in all three-post infection condition was observed in the HPAIV infected duck lung tissues (Figure 6)”;

5) Table 3 must be introduced at line 329.

Round 2

Reviewer 2 Report

Comments and Suggestions for Authors

The points I raised have generally been corrected.